

# Salinization origin of Souf Terminal Complex: Application of
# statistical modelling and WQI for groundwater management
Hafidha Khebizi[1], Bachir Benlaoukli[2], Foued Bouaicha[3], Patrick Adadzi[4] and Omar Bouras[5]
[1]MDWR laboratory, National Higher School of Hydraulics (NHSH), Blida, 09000, Algeria.
[2]MDWR laboratory, National Higher School of Hydraulics (NHSH), Blida, 09000, Algeria.
[3]Département de biologie appliquée.Université  frères Mentouri, 25000, Constantine , Algérie.
[4]Institute for Groundwater Studies, Faculty of Natural and Agricultural Sciences, University of the Free State Bloemfontein,
South Africa.
[5]WESD Laboratory, Department of Process Engineering - Saad Da'hlab-Blida University 1, Blida, 09000, Algeria.
*Correspondence to:*  Hafidha Khebizi (h.khebizi@ensh.dz)
**Abstract.** The natural salinization of Souf sandy Terminal Complex groundwater notably Pontian and Mio-Pliocene
has increased four times for the last 30 years, because of over pumping for drinking and irrigation. Application of the
statistical modelling using multivariate analysis, and the Water Quality Index, to evaluate the groundwater variables
have been done for groundwater management, by the investigation of water samples collected from 25 boreholes, in
May 2018. Cluster analysis identified three main water types based on the major ionic contents. Factor Analysis and
Principal Component Analysis methods confirmed the cluster analysis results.  The water groups have sodi-potassic
facies which dominate in the western part of Souf, compared to the eastern part and they have poor quality. An osmosis
phenomenon allowed the homogenization of Pontian and Mio-Pliocene groundwater. The contact of Terminal
Complex with the Eocene dolomite and Senonian evaporitic host rocks allows introducing a new preferential
dissolution corridors concept in which an underground leaching front occurs with the increased pumping. Overlying
sandy rocks subsidence can be produced gradually with a rise in the static groundwater level because of the leached
underground Senonian evaporitic rocks. Closure of wells intersecting the evaporitic layers and minimizing of pumping
from Terminal Complex groundwater in the Southwest part of Souf is strongly recommended, and the groundwater is
requiring treatment before supply.
**Keywords**: Terminal Complex. Statistical modelling. Groundwater management. Salinization. Souf.
**1 Introduction**
Souf arid climatic characteristics and its Erg geomorphology allow only a occasional appearance of water on the surface. The
presence of permanent saline areas as chott and sebkha that form the natural outlets of Terminal Complex groundwater
indicate that this groundwater is naturally saline. The increasing number of wells and pumping of the Terminal Complex
aquifer for the last 30 years has led to a hydrodynamic destabilization, and a four-fold increase of salinization levels in the
groundwater compared to (WHO, 2011) standards. Several authors (Guendouz A. and al. 1992, Moulla A. 2003, Tabouche
N. and al. 2004, Remini 2006, Habes S and al. 2016) discussed this problem in Souf region. In the Tunisian limitroph regions,
Tarki M  (2011) also investigated a similar phenomenon.
The objective of this research in addition to the previous investigations is to determine the impact of the host rock lithology
mineralization, which forms the source of salinization.  This study will enable the discussion for the first time of the following
three components:
1. the lithological evolution and the lateral passage of the host rock sedimentary formations and spatially explanation of the
various ions distribution in the groundwater, and the water chemical composition changes with the lithological variations
of the host rock. This allows an interpretation of the concordance between the water groups distribution and the different
host rock lithological natures by giving new mineralization corridors concept.





2. the osomosis phenomenon effect on the homogenization in the chemical composition of the Pontian and Mio-Pliocene
which form the Terminal Complex.
3. the hypothesis about the relation between the water salinization and the rising static groundwater level at a regional scale.
Statistical modelling and Water quality index (WQI) methods were integrated in this research to investigate water-rock
contact behavior in Souf Terminal Complex groundwater. The investigation of groundwater quality variables by the
determination of the different evaporitic minerals, which cause the water salinization, is a very significant contribution to
groundwater management in the region.
**1.1 Geographical setting**
Souf is an administrative entity formed by 18 municipalities which occupy the center of Oued Souf Wilaya. It is limited to
the North by Melgheir Chott depressions; to the West by Oued Righ; to the East by Chott Djerid and to the South and
Southeast by the Erg (Fig.1).
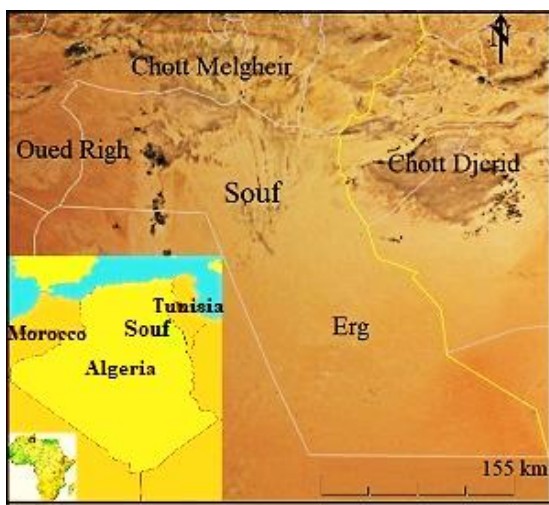

54                          **Figure 1**: Geographical setting of Souf (© google earth)

**1.2 Geological setting**
Souf belongs to the northern part of the lower Sahara basin (Fig. 2). The sandy Pontian and Pliocene formations outcrop in
the Western part and the Southwest towards Touggourt, and are formed by lacustrine limestones. The Upper Cretaceous is
formed by carbonate, and it just appears in the Southeast. Quaternary sandy formations cover Mio-Pliocene formations with
recent dunes as Sifs Soltane, El Yhoudi, in the West, and El Arif in the South; Erg as Bou Lossa, Bou Fegoussa, and Touil
in the South and Sahane as Bel Lefa, En Nsi, Deklat Chechili.



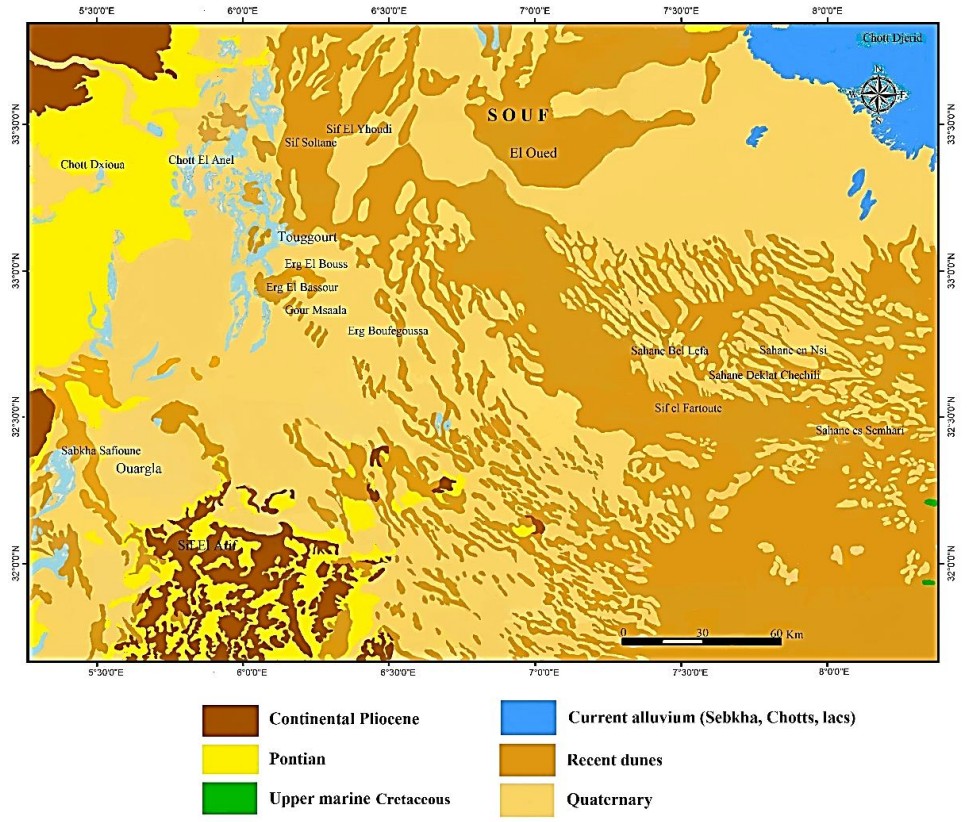


**Figure 2**: Geological map of Oued Souf (M.G. Betier 1952, modified).

Current alluviums are mainly presented by evaporitic formations notably the chotts as chott Djerid in the East, in the West
chotts of Dxioua, El Anal, Chegga, El Meryeir, El Melah, A. Rouma, and Sebkha as Safioune in the Southwest. Also, swamps,
dayas and gypsum-salt crusts occur at some places. At the regional scale, the petroleum stratigraphic logs (Sonatrach, 1965,
1975, 1985) revealed in-depth, a complete Cretaceous sedimentary series since the Neocomian (Fig.3).
The geology of Oued Souf shows that sedimentation is thicker in the center compared to that on the edges where lateral
lithological variations occur due to the paleogeographic context and the faulty structure. The basin may still be sagging in
the present day (Cornet, 1961). Neocomian is formed by clay with interactions of sandy limestones and dolomites. The lower
and middle Cenomanian are formed by clay-marl. The upper Cenomanian is made by limestone, dolomite and anhydrite, and
Turonian is formed by dolomite and limestone. Senonian lagoon type, in its lower part, is composed of dolomite with salt
and anhydrite pasts. The upper part is composed of limestone. Eocene is formed of dolomite. Pontian is formed by sand and
limestone where limestone thickness changes from the South to the North in its upper part. Mio-Pliocene is sandy



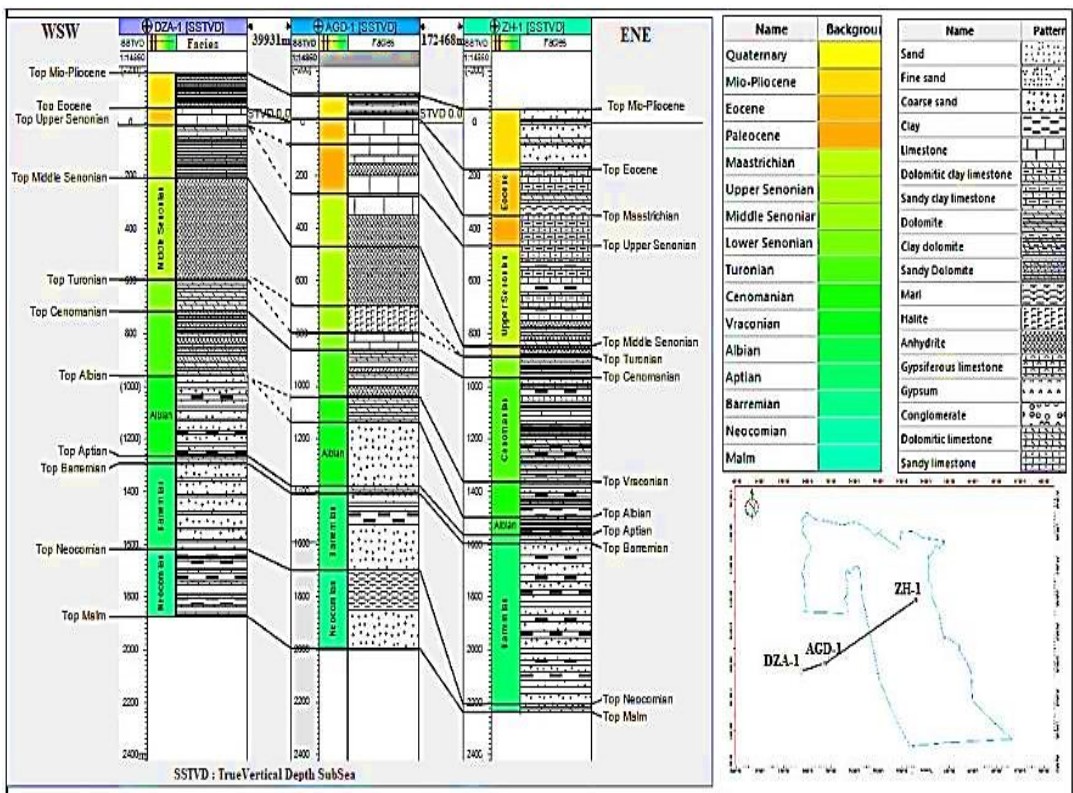


**Figure 3**: Lithostratigraphical log correlations

Eocene is the thinner formation (50m thick) comparing to the other formations due to its erosion. It disappears totally in
some places going toward the West and the South West of Souf where Mio-Pliocene sandy layers contact directly Senonian
evaporitic layers (Fig.3).
**1.3 Hydrogeology**
Terminal Continental (Didier Roger et al., 1969) or Terminal Complex (ERESS, 1972), is the name of aquifers found in
Senonian, Eocene and Mio-Pliocene geological formations which are interconnected and therefore belong to the same
groundwater, excepted chotts where the middle and upper Eocene is intercalated. Turonian aquifer is more individualized
because of lagoon Senonian impervious cover. Terminal Complex outcrops in chotts, in the eastern flank of Dahar and J.
Nafusa, in Tinrhert, in Tademait plateau and in M'zab Ridge (Fig. 4).



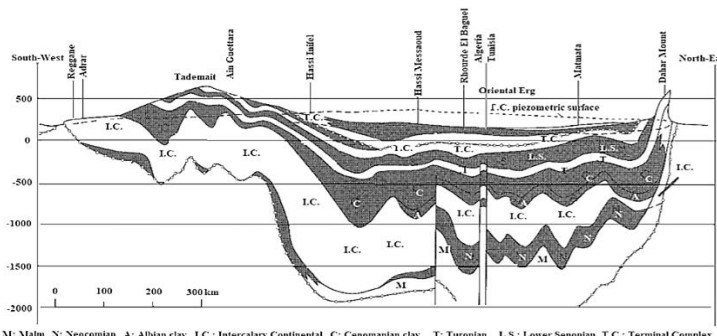


**Figure 4**: Geological section showing Terminal Complex in Sahara (UNSECO, 1972)
Terminal Complex groundwater is supplied directly by meteoric waters of lower Sahara artesian basin, by water flowing
through lower Sahara valleys infiltrating into subsoil along Wadi. Also, it is provided by water coming from the Saharan
Atlas and Wadi coming from Oued Igharghar in the South. Groundwater flow direction in Souf is from the West to the East
and from the South West to the Northeast (Fig. 5).

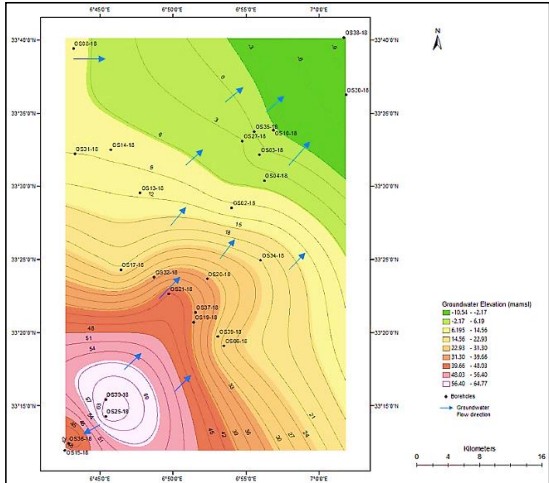


**Figure 5**: Map of Souf Terminal Complex groundwater flow direction
**2    Materials and methods**
**2.1 Sampling analysis**
A sampling campaign was organized from 30 April to 05 May 2018, in collaboration with Oued Souf Hydraulics Department
and the National Water Resources Agency (NWRA) of Touggourt branch where 25 water sample has been collected from
wells intended for drinking water supply in Souf. 07 samples were taken from Pontian groundwater (OS08-18, OS13-18,
OS20-18, OS21-18, OS33-18, OS34-18, OS35-18) and 18 samples from Mio-Pliocene groundwater (OS02-18, OS03-18,
OS04-18, OS06-18, OS10-18, OS14-18, OS15-18, OS17-18, OS19-18, OS25-18, OS27-18, OS30-18, OS31-18, OS32-18,
OS35-18, OS37-18, OS38-18, OS39-18). The temperature (T°c), the electrical conductivity (EC) and pH (hydrogen potential)

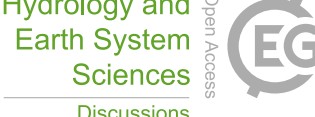

have been determined immediately after sampling using two portable devices: pH meter and conductivity meter (WTW). In
the laboratory, water samples were analyzed at the water treatment laboratory of NWRA Ouargla. Sulfates were measured
by turbidimetry at 495 nm wavelength. Calcium, sodium and potassium cations were determined by flame photometry.
Chlorite is measured by flame photometry. Nitrates were assayed by chlorimetry at 520 nm appropriate wavelength.

**2.2 Statistical modelling**

Variability of the Terminal Complex groundwater quality parameters are linked to numerous processes such as mineral
dissolution and precipitation, reverse ions exchange, osmosis phenomenon, and anthropogenic process. Multivariate
statistical analyses such as factor analysis (FA), principal component analysis (PCA) and hierarchical cluster analysis (HCA)
were applied to the standardized data set of ten (10) groundwater quality parameters (pH, EC, $Ca^{2+}$, $Mg^{2+}$, $Na^+$, $K^+$, $NO_3^-$,
$HCO_3^-$, $Cl^-$ and $SO_4^{2-}$), to elicit the hydrologic and biogeochemical processes affecting water quality. These techniques have
been successfully used by scientists on hydrochemistry to classify water (Francisco Sânchez-Martos, 2001; Guler et al. 2002;
Demirel and Guler 2006; Cloutier and al. 2008; Tenalem Ayenew and al. 2009; Belkhiri and al. 2010; Varol et al. 2012;
Salman et al. 2014; Murugesan Athimoolam and al. 2014; Subba Rao, 2014; Taqveem Ali Khan, 2015; Sarita Gajbhiye and
al., 2015; Lianne McLeod, 2017; Nabil Darwesh and al. 2019; Bouaicha et al. 2017). Cluster analysis is a useful tool for
hydrochemistry investigation to summarize all information by grouping water samples into separate significant groups in the
geologic and hydrologic context for a better understanding of the hydrogeochemical process occurring in the study area
(Guler et al. 2004; Tenalem Ayenew and al. 2009; Taqveem Ali Khan, 2015; Singh et al. 2017). It is done based on their
similarities by Q-mode HCA method on the normalized data set. Also, FA and PCA are widely used to reduce sets of
observations of many variables using associations between them (Bouaicha et al. 2019). The deduction is achieved by
diagonalization of the correlation matrix which obtains a new data set uncorrelated (orthogonal), arranged in a decreasing
order of importance named principal components (PCs) (Singh et al. 2004). In this study, PCA was carried out on the
standardized data sets and sorted using eigenvalues greater than one as these are considered significant influences towards
the hydro-geochemical processes (Semar and al., 2013). Varimax rotation was executed to these PCs to make the factors
easier to interpret according to the hydrochemical or anthropogenic processes controlling the groundwater quality.

**2.3 Water quality index**

The Water quality index WQI is a method given by Brown et al. (1972) which is a recognized technique that offers a useful
tool that simplifies the expression of water quality (Chauhan and Singh, 2010). It is a numerical expression where water
quality data set is summarized into simple terms (excellent, good, poor, etc.) There are numerous water quality indices (WQI)
developed by governmental agencies around the world. Authors have widely used this method (Amadi 2011; Gebrehiwot
and al. 2011; Desai and Desai 2012; Aly and al. 2014; Amaliya and Kumar 2015; Goher and al. 2015; Paul and al.; 2015;
Bouteraa and al. 2019). WQI value water quality status is mentioned in the table following.

**Table 1**: Water quality assessment as per weight arithmetic WQI method

|  | WQI value | Water quality status |
| --- | --- | --- |
| Excellent | <50 | Excellent |
| Good | 50-100 | Good |
| Poor | 100.1-200 | Poor |
| Very Poor | 200.1-300 | Very Poor |
| Unsuitable for irrigation purpose | >300 | Unsuitable for drinking purpose |





**3    Results and discussions**
**3.1 Hydrogeochemical process**
Cluster analysis has led to identifying the different chemical facies of the groundwater by the Q-mode HCA method. Sulfate,
chloride ions and electrical conductivity (EC) seem to be a determining factor in differentiating the different water groups
and indicate high salinity water (Fig. 6).

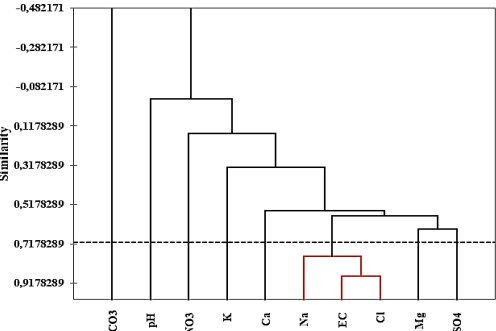


**Figure 6**: Dendrogram showing the hierarchical clusters of analyzed parameters
This method has led to identifying three water groups, which are compared with the World Health Organization (2011)
standards for water quality parameters (Table 2).
**Table 2**: Physico-chemical analysis results of Souf Terminal Complex groundwater

| | Group 1 | | | | Group 2 | | | | Group 3 | | | | WHO |
|---|---|---|---|---|---|---|---|---|---|---|---|---|---|
| | Min | Max | Mean | SD | Min | Max | Mean | SD | Min | Max | Mean | SD | 2011 |
| **pH** | 7,63 | 8,08 | 8,07 | 0,22 | 7,77 | 8,28 | 8,02 | 0,18 | 7,98 | 8,17 | 8,06 | 0,10 | 8.5 |
| **EC** | 4070,00 | 4080,00 | 4197,14 | 110,00 | 4510,00 | 5180,00 | 4912,50 | 295,91 | 4530,00 | 4600,00 | 4560,00 | 36,06 | 1500 |
| **Ca²⁺** | 232,50 | 235,00 | 240,89 | 10,77 | 247,30 | 317,50 | 269,98 | 22,43 | 247,50 | 262,50 | 257,50 | 8,66 | 75 |
| **Mg²⁺** | 151,00 | 204,80 | 177,00 | 19,86 | 153,60 | 235,50 | 203,84 | 24,20 | 192,00 | 204,50 | 197,87 | 6,29 | 50 |
| **Na⁺** | 400,00 | 460,00 | 431,96 | 14,62 | 450 | 580,00 | 496,87 | 50,47 | 462,00 | 510,00 | 490,67 | 25,32 | 200 |
| **K⁺** | 19,00 | 34,00 | 26,25 | 5,63 | 26,00 | 34,00 | 30,13 | 3,09 | 26,00 | 31,00 | 28,50 | 2,50 | 12 |
| **Cl⁻** | 650,00 | 712,50 | 691,07 | 18,62 | 712,50 | 1000,00 | 819,19 | 102,05 | 750,00 | 837,50 | 789,33 | 44,41 | 250 |
| **SO₄²⁻** | 1000,00 | 1187,50 | 1115,18 | 80,89 | 1187,50 | 1375,00 | 1281,00 | 70,58 | 1050,00 | 1162,50 | 1120,83 | 61,66 | 250 |
| **NO₃⁻** | 4,00 | 12,50 | 12,10 | 4,73 | 6,50 | 22,50 | 13,38 | 5,80 | 9,00 | 31,50 | 23,33 | 12,45 | 10 |
| **HCO₃⁻** | 186,05 | 195,20 | 201,29 | 7,37 | 183,00 | 201,30 | 192,15 | 7,47 | 189,10 | 198,25 | 193,17 | 4,66 | 120 |

The following figure shows the hierarchical clusters of analyzed water samples.

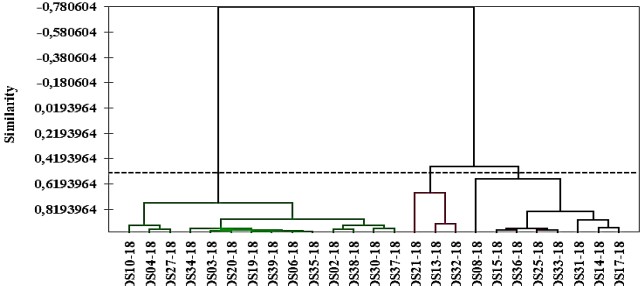


**Figure 7**: Dendrogram showing the hierarchical clusters of analyzed water samples



**Group 1** is formed by 14 wells in which 2 wells are from Pontian (OS20-18, OS34-18) and 12 wells from Mio-Pliocene
(OS02-18, OS03-18, OS04-18, OS06-18, OS10-18, OS19-18, OS27-18, OS30-18, OS35-18, OS37-18, OS38-18, OS39-18).
All wells are localized in the eastern part of Souf. The major ions abundance order is $Na^+ + K^+ > Ca^{2+} > Mg^{2+}$ and $SO_4^{2-} > Cl^-$
$> HCO_3^- > NO_3^-$. It exceeds four times the limit for drinking water standards (WHO, 2011). The hydrochemical type is sulfate
facies with $SO_4^{2-}$ (min = 1000,00 mg/l, max = 1187,50 mg/l, and mean = 1115,18 mg/l), and potassic with sodium (min =
400,00 mg/l, max = 460,00 mg/l, and mean = 431,96 mg/l). Bicarbonates exist with $Ca^{2+}$ (min = 232,50 mg/l, max = 317,50
mg/l, and mean = 240,89 mg/l).The concentrations of nitrate exceed the standards required for consumption in 10 wells in
which min = 11 mg/l and max = 22,5 mg/l.
**Group 2** is formed by 8 wells in which 2 wells are from Pontian (OS08-18 and OS33-18) and 6 wells from Mio-Pliocene
(OS10-18, OS15-18, OS17-18, OS25-18, OS31-18 and S36-18). The majority of wells are located in the West. The major
ions abundance order is $Na^+ + K^+ > Ca^{2+} > Mg^{2+}$ and $SO_4^{2-} > Cl^- > HCO_3^-$ and the hydrochemical type is characterized also by
a sulfate facies with $SO_4^{2-}$ (min = 1187,50 mg/l, max = 1375,00 mg/l, and mean = 1281,00 mg/l). Sodium is also dominant
with (min = 450mg/l, max = 580,00mg/l, and mean = 496,87mg/l). Bicarbonates exist with less importance with $Ca^{2+}$ (min
= 247,30mg/l, max = 272,5 mg/l, and mean = 269,98mg/l). $Mg^{2+}$ with (min = 153,60 mg/l, max = 235,50mg/l and mean =
203,84mg/l). Most samples exceeded four times the limit for drinking water norms (WHO 2011). The concentrations of
nitrate exceed the standards required for consumption in 06 wells in which min = 14 mg/l and max = 29,50 mg/l.
**Group 3** consists of three wells: 2 wells from Pontian (OS13-18, OS21-18) and one well from Mio-Pliocene (OS32-18).
These wells are localized in the eastern part of Souf. The major ions abundance order is the same and exceed four times the
limit for drinking water standards (WHO 2011). The hydrochemical type is sulfate facies with $SO_4^{2-}$ (min = 1050,00 mg/l,
max = 1162,5 mg/l and a mean =1120,83mg/l), and potassic with $Na^+$ (min = 462,00 mg/l, max = 510,00 mg/l, and mean =
490,67 mg/l). Bicarbonates exist with $Ca^{2+}$ (min = 247,50 mg/l, max = 262,50 mg/l, and mean = 257,50 mg/l).The
concentration of nitrate show that only OS13-18 exceeds the standards required for consumption with max = 31,50 mg/l.
Spatially, wells situated in the western part of Souf are more mineralized than those situated in the Est. These groups are the
most abundant on $SO_4^{2-}$, $Cl^-$ $Ca^{2+}$, $Mg^{2+}$ and $Na^+$. Piper diagram (Piper 1944) shows the potassic sulfate facies of these groups
(Fig. 8).

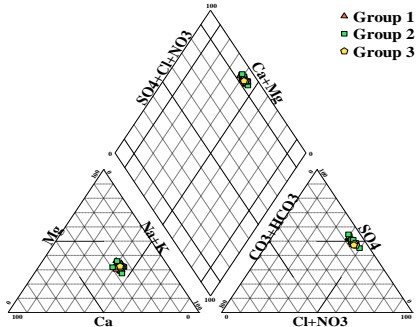

**Figure 8**: Piper diagram
Chadha diagram (Chadah, 1999) which is a modified version of the Piper diagram (Fig. 9) shows that most groundwater
samples are characterized by the dominance of alkaline ($Na^+$ and $K^+$) over alkaline earth ($Ca^{2+}$ and $Mg^{2+}$) and strong acids
($SO_4^{2-}$ and $Cl^-$) over weak acids ($HCO_3^-$).

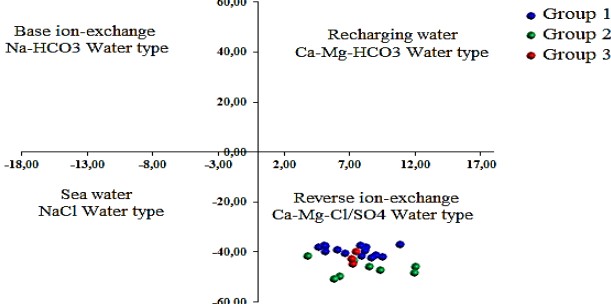

**Figure 09**: Chadah diagram
Most of the groundwater groups are situated in the field of reverse Ion-Exchange $Ca^{2+}$-$Mg^{2+}$-$Cl^-$/$SO_4^{2-}$ water type (Fig.9).



The correlation matrix results shown in Table 3 reveals an excellent correlation between the pairwise EC and $Cl^-$ (0.88), $SO_4^{2-}$
(0.71), $Na^+$ (0.73) and a good correlation with $Ca^{2+}$ (0.67) and $Mg^{2+}$ (0.65), indicating a strongly mineralized water. An
excellent correlation was revealed between $Na^+$ and $Cl^-$ (0.83) indicating the dissolution of halite. The correlation between
$Mg^{2+}$ and $Cl^-$ (0.62) indicates the dissolution of bischofite, and between $Mg^{2+}$ and $SO_4^{2-}$ (0.64) signifying the dissolution of
epsomite.

**Table 3**: Correlation matrix

| Variables | pH | EC | $Ca^{2+}$ | $Mg^{2+}$ | $Na^+$ | $K^+$ | $Cl^-$ | $SO_4^{2-}$ | $NO_3^-$ | $HCO_3^-$ |
|---|---|---|---|---|---|---|---|---|---|---|
| pH | **1,00** | | | | | | | | | |
| EC | 0,07 | **1,00** | | | | | | | | |
| $Ca^{2+}$ | 0,02 | **0,67** | **1,00** | | | | | | | |
| $Mg^{2+}$ | -0,12 | **0,65** | 0,51 | **1,00** | | | | | | |
| $Na^+$ | -0,10 | **0,73** | 0,49 | 0,51 | **1,00** | | | | | |
| $K^+$ | -0,01 | 0,35 | 0,42 | 0,42 | 0,30 | **1,00** | | | | |
| $Cl^-$ | 0,07 | **0,88** | 0,53 | **0,62** | **0,83** | 0,36 | **1,00** | | | |
| $SO_4^{2-}$ | -0,08 | **0,71** | 0,54 | **0,64** | 0,48 | 0,10 | 0,47 | **1,00** | | |
| $NO_3^-$ | -0,02 | 0,13 | 0,16 | 0,09 | 0,30 | 0,26 | 0,31 | -0,15 | **1,00** | |
| $HCO_3^-$ | -0,06 | -0,62 | -0,50 | -0,48 | -0,65 | -0,53 | -0,77 | -0,26 | -0,46 | **1,00** |

Factor analysis with varimax rotation applied to data has shown 63.85% of the total information, where PC1 represents
41.38%, and PC2 represents 22.27%. PC1 has a strong positive loading on electrical conductivity (EC), $Na^+$ and $Cl^-$, $Mg^{2+}$,
$SO_4^{2-}$, a moderately positive loading on $Ca^{2+}$, and a strong negative loading on $HCO_3^-$ indicating geogenic process in which
mineral dissolution, reverse ions exchange and osmosis phenomenon could intervene. While PC2 has a strong positive
loading on $NO_3^-$ indicating an anthropogenic process (Table 4).

**Table 4** Score of PCA after Varimax rotation

| | **PC1** | **PC2** |
|---|---|---|
| pH | -0,084 | 0,128 |
| EC | **0,898** | 0,258 |
| $Ca^{2+}$ | **0,708** | 0,251 |
| $Mg^{2+}$ | **0,795** | 0,142 |
| $Na^+$ | **0,701** | 0,445 |
| $K^+$ | 0,297 | 0,577 |
| $Cl^-$ | **0,755** | 0,508 |
| $SO_4^{2-}$ | **0,881** | -0,244 |
| $NO_3^-$ | -0,092 | **0,827** |
| $HCO_3^-$ | -0,508 | -0,74 |
| **% variability** | **41,381** | **22,476** |
| **% cumulated** | **41,381** | **63,857** |
| Interpretation of the process | 1. Mineral dissolution and/or precipitation<br>2. Cations exchange<br>3. Osmosis phenomenon | Anthropogenic pollution |

A scatter-plot (Fig. 11) of PC1 versus PC2 reveals that all water groups are well distinguished from each other in the PC
space and coherent with groupings extracted from Q-mode HCA.





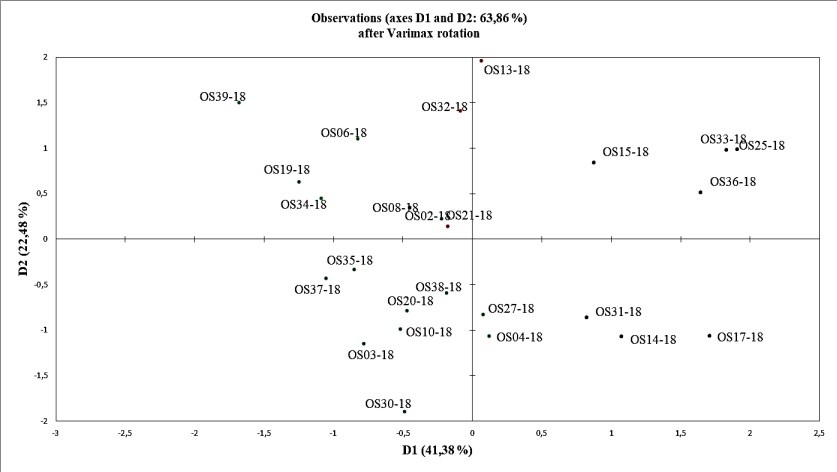


**Figure 10**: PCA biplot of water samples based on the first two axes

**3.2 Water quality index**
WQI method developed for groundwater parameters represents the overall quality of water according to its purity degree.
For the study area, WQI value was computed for drinking water using the guidelines of WHO (2011). The EC, pH, $Ca^{2+}$,
$Mg^{2+}$, $Na^+$, $K^+$, $Cl^-$, $SO_4^{2-}$, $HCO_3^-$, and $NO_3^-$ have been used to obtain the WQI. Results revealed that 23 wells had WQI poor
water quality and two wells (OS25-18, OS36-18) had very poor water quality (more than 200). WQI values for groundwater
samples are shown in Table 5.

**Table 5**: WQI of the Terminal Complex water groups

|         | WQI Value        | Quality            |
|---------|------------------|--------------------|
| Group 1 | 152,86 to 196,38 | Poor               |
| Group 2 | 162,63 to 209,08 | Poor to very Poor  |
| Group 3 | 164,46 to 192,18 | Poor               |


The spatial abundance of mineralization is due to the geology context of Terminal Complex groundwater. In the West and
Southwest part of Souf, the Eocene dolomite formation is very thin due to its erosion and it disappears totally going toward
the West where the sandy Pontian and Mio-Pliocene layers become in direct contact with Senonian evaporitic and salty
layers. In this region, the groundwater facies indicates the dissolution of sulfate and chloride evaporitic minerals, which are
found in the potassic lagoon deposit of Senonian evaporitic. The most abundant chloride is halite (NaCl). Dissolution of
halite reaction is as follow:

$$NaCl \rightarrow Na^+ + Cl^-$$

Other chlorides can be associated, such as bischofite (MgCl) and sylvine (KCl). Also, the most abundant sulfate is gypsum
($CaSO_4$) according to the following formula:

$$CaSO_4 \rightarrow Ca^{2+} + SO_4^{2-}$$

Other chlorides less abundant can be associated to gypsum such as epsomite ($MgSO_4$). In this case, water is enriched in
$SO_4^2$, $Cl^-$, $Ca^{2+}$, $Mg^{2+}$ and $Na^+$ and this is clearly observed in group 2 and 3. Toward the North, the East, and the center,
Eocene dolomite forms the roof of the sandy Pontian and/or Mio-Pliocene layers and dolomite (Ca, $MgCO_3$) dissolution is
more critical where water enrichment in $Ca^{2+}$, $Mg^{2+}$ and $HCO_3^-$ is significant and this is clearly observed in group 1. Dolomite
dissolution reaction is follow:



$$Ca, Mg\ (CO_3)_{2(dolomite)} + 2H_2O + 2CO_2 \rightarrow Ca^{2+} + Mg^{2+} + 4HCO^-_3$$
Senonian evaporitic minerals are the most soluble compared to carbonates (dolomite) due to their dissolution rates that vary:
0.03 and 0.05 $gm^2/s$ for gypsum, $3g/m^2/s$ for halite (Cubillas et al., 2005). Calcite has a rate of $10\text{-}4g/m^2/s$ (Cubillas et al.,
2005) and shows much more a precipitation tendency. The dissolution of the evaporitic minerals typically associated in
potassic deposit of the Senonian evaporitic due to the water-rock contact form the salinization source in the West and the
Southwest of Souf. A new concept of a preferential dissolution corridor may be introduced in this research for the first time
in the study area. It is mainly related to the host rock lithology. In case of Senonian evaporitic host rock, sulfates and chlorides
dissolution allow $Ca^{2+}$, $Mg^{2+}$, $Na^+$, $K^+$, $Cl^-$, $SO_4^{2-}$ enrichment. While, in case of dolomite host rock, $Ca^{2+}$, $Mg^{2+}$ and $HCO_3^-$
are much abundant in water and develop a carbonate dissolution corridor.
An osmosis phenomenon could intervene in the homogenization of Pontian and Mio-Pliocene groundwater mineralization in
group 2 and group 3. This mechanism allows ions circulation of the most concentrated waters in chemical elements towards
waters less rich in these elements through layers of Pontian clay roof, which is considered as a semi-permeable membrane.

*Salinization hydrochemical and hydrodynamic effect hypothesis*
The waters by their double role as erosive and transport agent enrich themselves in chemical elements simultaneously with a
preferential underground leaching mechanism of the host rock. The most soluble minerals are those of the Senonian evaporitic
host rock than those of Eocene dolomite host rock. The host rocks contain sulfates and chlorides occurring in abundance as
gypsum, anhydrite and halite, with less abundant sulfate and chloride evaporites such as epsomite, sylvine and bischofite
associated with the potassic lagoon deposit of Senonian. Thus, Senonian evaporitic host rock dissolution in the South and the
West part of Souf over time allows a significant departure of these minerals, which can be found again on the surface as
ephemeral minerals in the natural discharge zones of the groundwater (chott and sebkha). The underground leaching action
depends on water flow velocity and pumping rates, which generates in-depth vertical movements of water and allows the
creation of cavities. Under the load effect of the overlying sandy rocks, these cavities are filled, and lead to a gradual
lithological subsidence and the rise of the overlying sandy groundwater static levels. Authors have noted the joint dissolving
and subsidence problem (Benito G. 1995; Anthony H. 1999; Charola. 2007) which has dangerous consequences on buildings
(Bergeron C and al. 1983). In lower Sahara, this phenomenon is discussed for the first time in this research. It is not critically
observed and investigated because of sandy loose lithological nature of the groundwater and the dune masses that cover them.
The lithological subsidence may occur at regional scale gradually for few millimeters depth, on the favor of Senonian
evaporitic dissolution corridor. This action depends on the quantity of leached evaporitic minerals, recharge and discharge
groundwater periods and the increased groundwater pumping in these areas. (Fig.11).





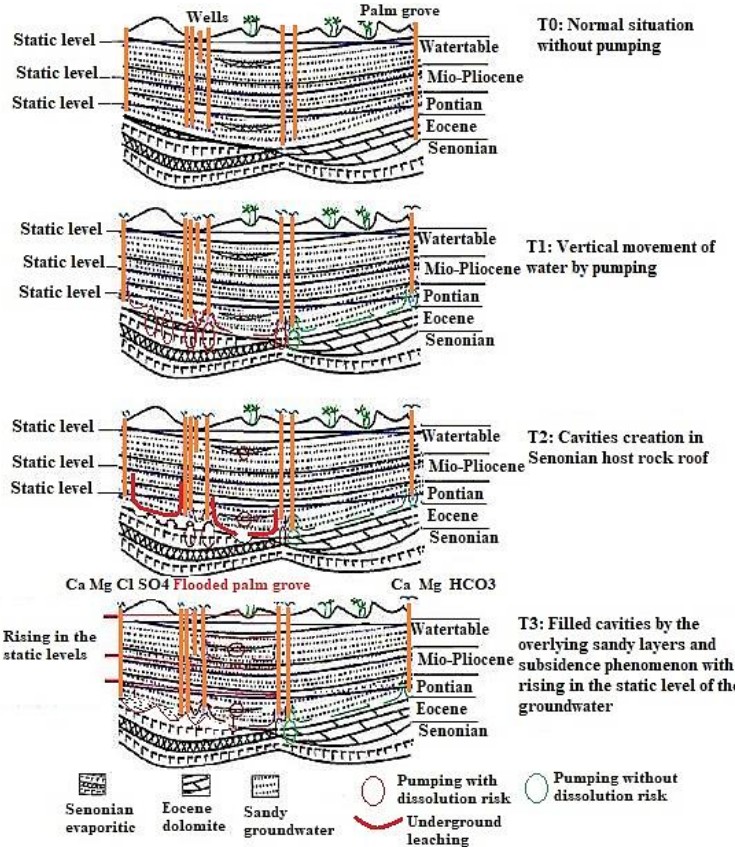


**Figure 11:** Hydrodynamic salinization effects scenarios

The current situation of static level of the watertable in Souf especially in Sidi Mestour zone could be a result of a gradual
subsidence of the underlying sandy layers of Terminal Complex about few millimeters after significant underground leaching
of the evaporitic minerals quantities (Fig 12).

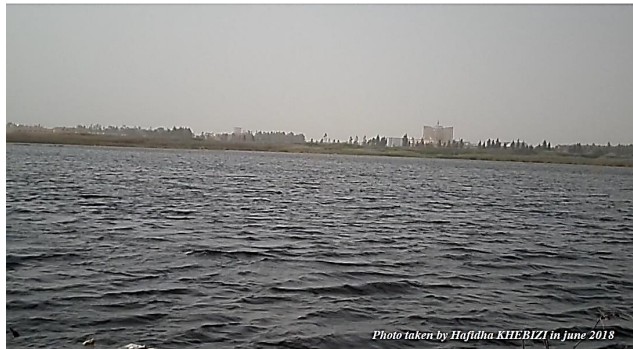


**Figure 12**: Photo showing the abnormal static level of the watertable in Sidi Mestour



## 4   Conclusion

The Sandy Terminal Complex groundwater in contact with carbonate Eocene and evaporitic Senonian causes the mineralization of water. During groundwater's recharge period, highly mineralized waters are leaching with chemical elements exceeding WHO (2011) norms. The dissolution of the dominant evaporitic minerals such as halite, gypsum, and anhydrite, and other associated evaporitic minerals of halite such as sylvite, epsomite and bischofite occurrences permitted enrichment of water in sulfate and chlorate. The water groups distinguished are enriched in mineralization according to the groundwater host rock.  The carbonate host rock showed less mineralization of sulfate and chlorate, while the evaporitic layers produced abundant elements of sulfate and chlorate. This allows the postulate in the presence of two different mineralization corridors. The first is located in the West and Southwest of Souf, following water flow direction and allowing enrichment in $Na^+$, $K^+$, $Mg^{2+}$, $Cl^-$ and $SO_4^{2-}$ where groundwater host rock is evaporitic Senonian. The second corridor is located in the North part of Souf. It promotes enrichment in $Ca^2$, $Mg^{2+}$ and $HCO_3^-$, with a host rock of limestone and dolomite Eocene. An osmosis phenomenon may intervene to homogenize the mineralization of Pontian and Mio-Pliocene groundwater. This mechanism allows ions circulation of the most concentrated waters in chemical elements towards waters with less enrichment through layers of Pontian clay roof, which is considered as a semi-permeable membrane. The interaction of the groundwater with Senonian evaporitic layers is regarded as subterranean preferential leaching, that was accelerated with pumping rates, and risks inducing the gradual subsidence of the overlying sandy layers, and rising static levels of the groundwater and acceleration of the dissolution-subsidence cycle. Further research and investigation are recommended:

- to delimit areas where groundwater is in direct contact with the evaporitic and salty Senonian layers.
- to identify and prohibit over-pumping in areas of high dissolution risk.
- for identification of appropriate water treatment method before supply and utilization.

Other multidisciplinary work is strongly recommended especially the geophysical study, to understand the groundwater and the host rock structure for sustainable groundwater management.

**Authors contribution**

Ms Khebizi H. did the sampling campaign and the various analysis. She did the geological study and the interpretation of data analysis using the statistical modelling and WQI. She discussed for the first time the concept of preferential dissolution corridors and introduced the phenomenon of osmosis in the groundwater mineralization. She discussed also for the first time the relation between the salinization of the Terminal Complex and the phenomenon of the watertable rising static level. Dr. Benlaoukli B. contributed in the geological study. Dr. Bouaicha F. helped in the statistical analysis and WQI calculation. Mr. Adadzi P. helped in the groundwater flow mapping and the redaction of the article and Pr. Bouras O. helped in the interpretation of water-rock behaviour.

**Code/data availability**

In this work, Excelstat software was used for the statistical modelling. For data availability, sampling campaign measurements and laboratory analysis of 25 water samples taken in 2018 are available in the supplement document with their geographic coordinates.





**Competing interests**

The authors declare that they have no conflict of interest.

**Acknowledgements**

I would like to thank the antenna of the National Water Resources Agency of Touggourt and the Hydraulic Department of El Oued for their support during the carrying out of the sampling campaign.

**Formatting of funding sources**

This research did not receive any specific grant from funding agencies in the public, commercial, or not-for-profit sectors.

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

45.