# Peer review of "Salinization origin of Souf Terminal Complex: Application of 1 statistical modelling and WQI for groundwater management 2"

_Hydrology and Earth System Sciences, 2020_

## Referee Comment (RC1) · Anonymous Referee #1 · 8 Oct 2020

This is the review for the manuscript "Salinization origin of Souf Terminal Complex: Application of statistical modelling and WQI for groundwater management". The authors present an investigation on the hydro-geochemical data in Souf region of northern Algeria. They have used very trivial instruments to analyze the water quality data collected within a week in 2018. They have also presented some statistical analyses without discussing the physical significance of employing them. I have strong doubts on the data and analyses presented. Therefore, I don't find it suitable to publish in HESS. I have the following major comments:

1. The water quality samples are collected between April 30 and May 5 of 2018 from

25 locations. The samples are collected during summer time. Salinization might be common during this time. The authors need to use additional data during different seasons and compare the estimate.

2. "Sulfates were measured by turbidimetry at 495 nm wavelength. Calcium, sodium and potassium cations were determined by flame photometry. Chlorite is measured by flame photometry. Nitrates were assayed by chlorimetry at 520 nm appropriate wavelength." [Lines 102-104] The instrumentation used for estimating the water quality parameters are very trivial with high uncertainty comparing the modern instruments. I have doubts on the water quality characterization using the data from these instruments as they are very similar (Figure 8).

3. Section 2.3: Water quality index: Details are needed here. How did the authors compute water quality index in this study?

4. "Section 3.1" Discuss the physical significance of the cluster analysis and discuss the results here. The authors have made three different groups based on cluster analysis, however, multiple overlaps are present in the data (Figure 8). How to distinguish between two different group's sample using this analysis? If the rationales are missing, cluster analysis only looks a mathematical tools nothing else.

5. "Table 2: Physico-chemical analysis results of Souf Terminal Complex groundwater". Please mention the units of each parameters. I believe the value is in decimal. If yes, remove the comma.

6. The cluster analysis based grouping has strong overlap, which is also visible in water quality index estimates (Table 5). Again, physical significance is missing here.

7. Conclusion: "The dissolution of the dominant evaporitic minerals such as halite, gypsum, and anhydrite, and other associated evaporitic minerals of halite such as sylvite, epsomite and bischofite occurrences permitted enrichment of water in sulfate and chlorate". There are no such analysis present in the manuscript to claim this. Authors are

concluding based on qualitative data.

8. Conclusion: "The water groups distinguished are enriched in mineralization according to the groundwater host rock. The carbonate host rock showed less mineralization of sulfate and chlorate, while the evaporitic layers produced abundant elements of sulfate and chlorate. This allows the postulate in the presence of two different mineralization corridors." Authors need to provide more analyses before claiming this here.

9. Conclusion: "An osmosis phenomenon may intervene to homogenize the mineralization of Pontian and Mio-Pliocene groundwater. This mechanism allows ions circulation of the most concentrated waters in chemical elements towards waters with less enrichment through layers of Pontian clay roof, which is considered as a semi-permeable membrane. The interaction of the groundwater with Senonian evaporitic layers is regarded as subterranean preferential leaching, that was accelerated with pumping rates, and risks inducing the gradual subsidence of the overlying sandy layers, and rising static levels of the groundwater and acceleration of the dissolution-subsidence cycle." Further analyses are needed to establish these facts. Authors should not claim these using limited one season data resource.

10. Improve the quality of the figures. Either the resolution is too low or the fonts are not visible for all of the figures.

11. The title should include 'Algeria', as the readers are not familiar with the region.

---

## Author Comment (AC1) · 14 Oct 2020

Dear referee (RC1), For your decision, I am completely against even for the remarks and I see that they are made technically after a superficial reading without making the relation between the various parameters introduced (in particular, the geographical, geological and the hydrogeological context and the new concept introduced). From your comments, I am not sure that you are hydrogeologist or hydrologist. That is why I'll give you more explanations to enrich your understanding of the problem discussed in this paper and its context and objectives. Answers to your comments are as follows: 1. First, you did a serious location error and yet it is very clear and easy to

make a geographic location using the map. The study area is not located in the North of Algeria. It is located in the northern part of the Sahara. There is a big difference between Sahara and the North of Algeria from geographical, geomorphological and geological context point of view. For the instruments used: Sorry I do not understand If it you are speaking about pH, conductivity, temperature measuring instruments. I think these are universal devices used by hydrogeologist and hydraulic specialists and large laboratories specializing in water sampling and analysis. Likewise, for the methods of hydrochemical analysis of the various cations and anions. These are universal analysis methods according to the universal standards used by the most researchers who published in journals of high scientific reputation. For doubts concerning the reliability of the analysis, it is necessarily to indicate that the ionic balance on all the samples analyzed is less than 5% with R2 equal to 0.71 which means that the quality of the analysis is very good. 2. The samples were collected over a week: Technically a week is the duration of a sampling campaign and I don't think a sampling campaign requires more. From a hydological, hydrogeological and even climatological point of view, this time is the period of the aquifer recharge threshold (high water period). You should note that in a desert area, it is the best sampling period which corresponds to the zone of the groundwater recharge threshold, of which all the climatic and hydrological conditions to carry out the mission are insured in particular the absence of the sand wind which prevents the mission and the specialists of desert area know that. From a sampling point of view, there are many studies that have been done cited as references that can give some idea about the chemistry and quality of the water. The selective sampling done during this mission that I did myself, concerns a well delimited area and it is indeed the center of the region of Oued Souf with an area of 400km2. We purposely followed the trace of the Oued to see on its two parts (East and west) the hydrochemical behavior on the basis of an excellent geological knowledge of the groundwater host rock. Also, you spoke of sampling during the summer! the sampling campaigns are not limited to seasons, there are two distinct periods (high water period and low water period) and the month of May corresponds to the threshold of the high

water period in the Saharan areas. 3. For the calculation of WQI, in the additional excel document, there is the formula used and we can introduce this formula in the text if it gives more understanding. 4. For the statistical methods, it is clear that the groups have been classified according to their hydrochemical similarities regarding the dominance of the major canions and anions. So this classification should have exactly a geographical distribution significance of the water samples according to the geological interpretation of the host rock. For this, it is useful to pay attention that samples of groups 2 and 03 are the most mineralized, and are found in the western and southern part of Oued and therefore the significance is much more geographical than anything else. This geographical distribution is directly related to the evaporitic nature of host rocks (Senonian) in the South while towards the east the host is much more carbonated (Eocene dolomite) where group 1 presents a significant values of $Ca^{2+}$, $Mg^{2+}$ and $HCO^{-3}$. For Piper diagram, there is no contradiction, if all the samples have a sulphate sodium-potassium character. This is due to the mineralization caused by the rock-water contact, and since the direction of water flow occurs from the Southwest to the Northeast, the impact of the enrichment in sulphated elements occurs much further in the south where the host rock is evaporitic. On the other hand, going north and east, the dolomitic hoste rock influences in addition to the sulphated mineralized water coming from the southwest. 5. For the unit, the major cations and anions are in mg / l. For the EC, it is per $\mu$s/ cm. 6. The WQI gives the degree of water potability. You should note that the hydrochemical analysis showed a mineralized water four times more than the standards of the WHO (2011) and this study aims to determine the origin of the excessive mineralization (salinity) and to make a zoning depending on the lithology of the groundwater host rock and therefore there is no overlap since the water is salty. 7. Dissolution of the dominant evaporitic minerals such as halite, gypsum and anhydrite, and other associated evaporitic minerals of halite such as the occurrences of sylvite, epsomite and bischofite allowed the enrichment of the water in sulfate and chlorate ". This is interpreted on the one hand through the significant values of the correlation matrix, between cations and anions (values close to 1) and on the other

hand through the interpretation of the sedimentological context of the evaporitic host rocks. It should be noted that the typical lagoon chemical sedimentation of the Senonian evaporites which allows the formation of the various minerals mentioned, is known in the potassium deposits of the Senonian formations of the Lower Sahara zone which has already been discussed by specialists in evaporitic rocks including the most important work of John K. Warren (2006). 8. Conclusion: "The water groups distinguished are enriched in mineralization according to the groundwater host rock. The carbonate host rock showed less mineralization of sulfate and chlorate, while the evaporitic layers produced abundant elements of sulfate and chlorate. This allows the postulate in the presence of two different mineralization corridors. I think in an academic study based on field data we can not claim and hypothesis are based on the geological and hydrogeological hydrogeochemical arguments. For rock-water behavior, if you cannot grasp this assumption, you just follow the logical interpretation given as follows: An aquifer in which the host rocks changes in lithology laterally is characterized by mineralization that varies according to the lateral passage of the groundwater host rock facies. If the host rock is evaporitic we should have a mineralization rich in chlorine, sulfate and manesian element. In case of carbonate host rock, we will find chemical elements dominated by calcium, magnesium and HCO3, by the effect of the contained minerals dissolution. The hypothesis of preferential dissolution corridors is not given from the imagination, but it is based on a simple underground mapping of the host rock lithology which favors this mineralization. Since evaporitic minerals have the most solubility in water compared to carbonates. A subterranean leaching occurs much more by the contact of water with the evaporitic rocks. The different dissolution zones were called dissolution corridors according to the chemical properties of the host rock minerals and their dissolution rate. 9. Conclusion: - "An osmosis phenomenon is a natural phenomenon that you can find in any two solutions of different ionic concentrations which are separated by a semi-permeable membrane. In our case, it occurs to homogenize the mineralization of Pontian and Mio-Pliocene groundwater. This mechanism allows ions circulation of the most concentrated waters in chemical elements towards waters

with less enrichment through layers of Pontian clay roof, which is considered as a semi-permeable membrane. - The interaction of the groundwater with Senonian evaporitic layers is regarded as subterranean preferential leaching. This action was accelerated with pumping rates, and risks inducing the gradual subsidence of the overlying sandy layers, and rising static levels of the groundwater by acceleration of the dissolution-subsidence cycle. You should note that water has a double action, it is an erosion agent and transport agent and therefore any solid evaporitic mineral which occupies a certain volume in the rock will leave a vacuum after its dissolution and the enormous quantities of minerals dissolved in water and besides which reappear again in the form of ephemeral minerals in the chotts (Merouane and Melgheir), will leave voids which are filled by the sands of the overlying layers. this induces a gradual subsidence, which cannot be seen due to the nature of the loose sandy desert material. Any subsidence creates by gravity effect a rise in the static level of the aquifer. For this point, this interpretation is not limited to the season, it is much more related to the rock-water behavior in the two different cases: evaporitic host rock and dolomite host rock. 10. We can add Algeria to the title. 11. We can improve more the figure quality.

Hope my explanation helped getting you back on track.

Regards.

---

## Short Comment (SC1) · 10 Nov 2020

The new concept of the preferential dissolution corridors introduced, for the first time in the Lower Sahara aims to better interprete the Terminal Complex sandy groundwater salinization in Souf and the implications on the overlying aquifers. This new concept can be a real progress in the understanding of the underground hydrodynamics and hydrochemistry. It is supported by a good understanding of the climatological, the geo-morphological and the geological context of the region on the basis of a selective water sampling of a well-defined area and allows in the future to: 1- better understand the relationship between the water salinization, the hydrodynamics of the groundwater and

the different geomorphological saharan aspects, in particular the fossil wadis (subsurface groundwater) and the chotts and sebkha. 2- better understand the implications of the subterranean dissolutions of evaporitic and salty rocks, on soil and also on the rise in the static level of the watertable. 3- develop a new strategy for a better management of water resources by introducing a new drilling programmes for drinking water supply and irrigation. This new concept is only the beginning of a new approach research in the region for the interpretation of underground hydrodynamics and hydrochemistry. A multidisciplinary geological, structural, geophysical studies and study of fluid mechanics, microbiology and other sciences would be essential in the future to benefit from this new concept and to better understand similar natural phenomena in other areas in the world.

---

## Short Comment (SC2) · 26 Dec 2020

During the last 30 years, the region of El-Oued knew an upwelling in the water levels of the surface water table. The consequences of this phenomenon are fatal for the man and the environment. We note the flooding of palm groves, the pollution of the surface water table, the deterioration of the living environment of the inhabitants and the progressive change in land use. The authors have applied the statistical modelling using multivariate analysis, and the Water Quality Index, to evaluate the groundwater variables by the investigation of water samples collected from 25 boreholes, in 16 May 2018. In their conclusion the authors have strongly recommended to close the wells

that intersect the evaporitic layers and minimize the pumping of groundwater from the terminal complex in the southwestern part of the Souf, as the groundwater requires pretreatment before supply. In my opinion, the methods proposed in the paper are questionable (multivariate analysis for limited sample) and the reported validation is far too weak for convincing the reader that they are reliable (the size of the sample). Compared to the doctoral thesis of BOUSELSAL Boualem, publicly defended at the University of Annaba (Algeria) in 2016, the article did not present any novelty or solution to existing problems. The article is an extended summary of the doctoral thesis of BOUSELSAL Boualem. BOUSELSAL Boualem in his thesis performed three measurement companies. The authors of the article performed only one company with results similar to the doctoral thesis. The rest of the work is well inspired and translated from the thesis.

You will find attached the PhD thesis of BOUSELSAL Boualem.

https://biblio.univ-annaba.dz/wp-content/uploads/2019/07/These-Bouselsal-Boualem.pdf

---

## Short Comment (SC3) · 29 Dec 2020

Is there any difference with the work already done by the OSS and the Swiss engineering office in the region regarding the problem of the groundwater upwelling and its impact on the environment and agriculture?
* * *

---

## Author Comment (AC2) · 1 Jan 2021

Response to the comment (SC2) of Mr. Bouselsal Boualem His attached file: BOUSELSAL Boualem PhD thesis https://biblio.univ-annaba.dz/wp-content/uploads/2019/07/These-Bouselsal-Boualem.pdf

Mr BOUSELSAL Boualem, First, I should thank you for the effort you did to study the problem of the watertable static level rising in Oued Souf although the comparison you made in your comment is not objective, since this watertable is not the object of my article presented for discussion. The remarks and observations cited below have just been listed to draw the attention of the scientific community to the different analysis

and interpretation methodology that I followed and the quality of the data that I used. My objective is to make a clear comparison between my article which constitutes a part of my current PhD thesis and your thesis which you have just presented in full for comparison! 1- I must point out that the unique common element between your entire thesis work and the part of my thesis expressed in this article is the study area which is Oued Souf particularly Souf area which occupies the center of El Oued Souf. 2- Your theme is focused on the watertable while my subject is related to the Pantian and Mio-Pliocene sandy groundwater (Terminal Complex). 3- You have mentioned the Terminal Complex only through the literature or old hydrochemical data carried out and already published in scientific articles of other researchers. For this, I would point out that you have used old analysis without specifying which groundwater in Terminal Complex it is, to determine the hydrochemical facies. Also you have not brought out the surrounding geological formations which are the origin of this mineralization by the contact rock-water. 4- Your own description "During the last 30 years, the region of El-Oued knew an upwelling in the water levels of . . . . . . . .. . .. land use" is related to the impact of the rise in the static level of the watertable. As I mentioned before this is your study object and not my objective in this paper. 5- The water sampling campaign that I carried out myself concerns the Pontian and Mio-Pliocene groundwater (Terminal Complex) and not the watertable. This campaign was done from April 30 to May 05, 2018 and I don't know in which paper did you read May 16? Regarding your remark relating to the number of samples, I draw the attention of the scientific community that this is a selective sampling limited to Souf area and not to the entire Oued Souf. Also, this is not only to bring out the hydrochemical facies that has already been treated by researchers cited in my reference. While, the novelty of my work for this recent campaign, in addition to updating the information on water quality, is the highlighting of the geological context role influencing the mineralization, based on a careful description of the host rock lateral evolution. This only appears in my work through the objective interpretation of the lithostratigraphic correlation study explaining the lateral passage of sedimentary formations. For this, I mentioned the influence of the Eocene

host rock and the Senonian host rock. Although you have ignored this main argument that I have used. I urge you to re-read the text and try to fully understand my methodology of analysis not only hydrogeological and hydrochemical or statistical analysis, but also geological and structural interpretation. I draw the attention of the scientific community that the new concept of preferential dissolution corridors is the result of a good understanding of both geological, structural, hydrogeological and hydrochemical context. For this, I hope allowing me to mention that my experience as sedimentology engineer and structural geologist magister and my current PhD thesis allowed me to better understand the underground hydrodynamics and hydrochemistry and to present this new concept in the Lower Sahara region. 6- In your thesis work, you just used a single lithostratigraphic section exceeding 250m synthesizing the sedimentary basin. The other logs do not exceed 200m which confirms the limitation of your interpretation to this depth. While, the lithostratigraphic logs presented in my work are used to establish lithostratigraphic correlations by the Petrel geological modeling software with the collaboration of geophysical specialists and structural geologists from the National Hydrocarbon Valorization Agency ( ALNAFT). The depth reached in these logs exceeds 700m. This objective is to highlight the concordance between the results of hydrochemical and statistical analysis and the geological interpretation for the determination of the Terminal Complex salinization origin in Souf. In addition, the logs used in my work aim to make lithostratigraphic correlations in order to see the lateral passage of the Terminal Complex host rock, in a regional scale, to clearly highlight the impact of the Eocene and Senonian on the mineralization of the Pantian and Mio-Pliocene groundwater. 7- The novelty of my work, which I cited above, is explained by highlighting the logical relationship between the distribution of water groups and the lithological nature of the host rock determined by the lithostratigraphic correlations. 8- This new concept of preferential dissolution corridors has never been discussed in another paper to explain the groundwater salinity and the impact of the Senonian evaporitic rocks dissolution on the overlying groundwater static level elevation in the Saharan areas and which you have completely ignored! Therefore, it is clear that my interpretation is completely different

from yours to explain the rise in the static level of the watertable. For this, I draw the attention of the scientific community that the watertable static level rising problem has been included for the first time in my article as one of the main results of the new concept introduced. 9- In page 117 of your thesis, you wrote a paragraph under the title of Temperature: "The aquifer of the Terminal Complex (CT) is located between the depths 120 m and 700m (limestone groundwater which is located between the depths 250m and 700m, is not exploited because of the high salinity)". I do not know which carbonate groundwater you are talking about. I think you should specify this limestone layer, its age, depth and thickness. I would like to inform you that the stratigraphic logs of Sonatrach have shown that the carbonate layer is even less than 100m and I specify in this case the Eocene which changes thickness laterally because of its erosion until its complete disappearance. In this case Mio-Pliocene sandy layer is discordant on the Senonian sometimes carbonate (upper Senonian) and sometimes lagoon evaporitic (middle Senonian) according to the continuity of the members of Senonian. This obliges me to encourage you to work in collaboration with structural geologists to fully understand the geological context of the Terminal Complex. I should mention that the lithostratigraphic correlation presented in figure 03 of my article is made by Petrel software as part of my thesis in order to follow the lateral lithological variations. It is the first step in bringing to light the new concept of preferential dissolution corridors that you have completely ignored! I would like to inform the scientific community that a great work of Terminal Complex geological modeling is underway using Petrel software, with collaboration of Alnaft specialists in order to highlight the impact of this new concept at a regional scale. 10- Finally, you said that the carbonate groundwater is not exploited! Which carbonate groundwater are you talking about Eocene or Senonian since you do not make a distinction. I can confirm to you that they are exploited although they are salty and you can check the ANRH Ouargla wells inventory. It is for this reason that there are small local desalination plants. You can contact the Directorate of Hydraulics of El Oued for more information. I hope that my comment and my explanation are more than enough to get you back on track. I think the scientific community can be convinced

by this detail. I found necessary to present even an idea on a work underway to show the importance not only of the new concept in research, but its impact on the obligation to make a new management strategy for the Terminal Complex shared by three countries. Regards.

---

## Referee Comment (RC2) · Anonymous Referee #2 · 19 Jan 2021

The manuscript evaluates the processes that drive the salinization in the Souf Terminal Complex. Both the scientific aspect and the presentation of the manuscript need to be improved before it can be considered for publication.

Major comments:

The research questions are generally well defined, but they also need to be directly addressed by the analysis presented in the manuscript. For example, how is the effect of osmosis quantified here and how is the cause-effect relationship between osmosis and groundwater quality established? What is the quantitative linkage between salinization and groundwater level?

The introduction should review the state of the art regarding the study of the linkage between groundwater quality and the processes of interest (e.g., lithological evolution, osmosis, anthropogenic withdrawal and the associated groundwater level change). The method section should be directly tied to the research questions and it should clarify the specific research questions that each method (e.g., cluster and PCA analysis) and metric (e.g., WQI) aim to address. Since the research questions are generally well defined, I suggest the authors organize the results and discussion tightly around each of the questions such that the readers are clear about how the scientific questions are addressed by the technical results. In addition, the discussion should be directly tied to the results presented in the manuscript.

Detailed comments:

Line 92. Please clarify the data source of Fig. 5.

Line 107. Note that here is already in the Method section, so I suggest the authors list only the processes that will be investigated here; more general descriptions/motivations should be moved to the introduction. Many processes are listed here, e.g., mineral dissolution, precipitation, reverse ions exchange, osmosis phenomenon, and anthropogenic process. Which ones are the focus of the manuscript? In the method section, it is also important to clarify how each of those processes is quantified and evaluated in the manuscript.

Line 109. Please briefly clarify what "standardized" refers to.

Section 2.3 Please clarify the rationale of employing WQI in this study and briefly clarify how WQI is calculated in this study.

Section 3. Before showing the findings from the statistical analysis, the measurements from the sampling analysis should be shown and the uncertainties of these measurements should be clarified.

Line 236. How is osmosis phenomenon quantified here and how is it linked to the result

presented in the manuscript?

Line 240-256. This reads quite general and is not directly based on the results presented in the study. Discussion should be directly tied to the results shown in the manuscript and please reference results/figures that support the argument in the discussion.

Fig 12. The figure itself does not show "abnormal" condition as it does not clarify what a normal condition would look like.

---

## Author Comment (AC3) · 13 Feb 2021

Dear referee (RC2),

First, I should thank you for your precious remarks, which reflect your real interest in the new concept introduced despite the fact that you suggest to improve the presentation of the manuscript publication. I took a lot of time to discuss your remarks with the co-authors and I hope you will find more explanation in this response. Of course, I will take into account your remarks, which are very useful but some of them need just more details to clarify my idea.

For your first remark, as you know osmosis, which is a physical diffusion phenomenon, occurs when two solutions of different concentrations are placed on either side of a "semipermeable" membrane. The concentration difference causes a difference osmotic pressure, which causes the water to move through the membrane and a dilution of the most concentrated solution. The water diffuses in both sens but the most important flow (thus the net flow) takes place towards the solution more concentrated.

a- Case of Pontian groundwater Pontian groundwater hydrochemistry can be distinguished into two types as mentioned before (More concentrated zone and less concentrated zone) according to the host rock. A difference osmotic pressure occurs to dilute the most concentrated water. This can be observed in the $Ca^{2+}$ and $Mg^{2+}$ anions concentration in OS34-18 with respectively 240mg/l and 204,8mg/l compared the other water samples of group1 that are more enriched. For group 2, less concentration of $Ca^{2+}$ and $Mg^{2+}$ is mentioned in OS08-18 with respectively 205 mg/l and 153,6 mg/l. For $Cl^-$ and $SO_4^{2-}$ anions, OS21-18 presents the less concentrations with respectively 750 mg/l and 1050 mg/l. b- Case of Mio-Pliocene groundwater OS03-18 is the less concentrated on $Ca^{2+}$ and $Mg^{2+}$ with respectively 132,5 mg/l and 163,8 mg/l. For $Cl^-$ and $SO_4^{2-}$ anions, OS39-18 presents less concentration with respectively 700 mg/l and 1000 mg/l. In the two cases, it seems that water samples taken for analysis are situated in the limit zone of the lateral passage in the host rock from carbonate Eocene to evaporitic Senonian. In these zones the difference in the chemical element concentrations allows the osmosis phenomenon.

The osmosis formula ($\pi$=RT C/M) used by Vant'Hoff that depends on the pressure of the water is not used in this work. Our objective is limited to the qualitative interpretation of water chemistry by relating the concentration of the water samples to their geological context. For example, in the case of the dolomitic host rock, in the initial state (Time 0) at a point A: the mass concentration of $Ca^{2+}$ and $Mg^{2+}$ in water is due to the dissolution of dolomite by rock-water contact according to the formula Ca, Mg (CO3)2 (dolomite) + 2H2O + 2CO2 → Ca2+ Mg2+ 4HCO-3. The mineralization

includes much more Ca2+, Mg2+ and HCO-3. After a certain time (T1) and during the permanent underground circulation of water, other chemical elements appear in the water in the same point A such as Na+, K+, Cl- and SO42-. These chemical elements come from the dissolution of other evaporitic leached minerals as NaCl, KCl, MgCl, CaSO4 and MgSO4. These elements are not very dominant in point A. In the zones of the lateral lithological passage (transition zone) of the dolomitic host rock toward an evaporitic rock (Point B) the pressure difference is observed. It results by ion concentrations differences which are greater in Point B than those at point A and less important in water sample taken in the area where the host rock is evaporitic (Point D). The phenomenon of osmosis occurs in the the transition zone or the in the limit of the preferential dissolution corridors areas where the pressure difference is remarkable and intervenes for the homogenization in the chemical composition of the Pontian and Mio-Pliocene groundwater. For the quantitative linkage between the salinization and the groundwater level variation, it is important to mention the effect of the volume occupied by the evaporitic and saliferous minerals in the solid state. This volume on the scale of the host rock reflects the roof thickness, which is in direct contact with the groundwater. The permanent Water-rock contact allows the mineral dissolution and their departure, which make vacuoles in the contact zone. By the load of sand subsidence can occur simultaneously with a rise in the groundwater static level. For the introduction, I should clarify that the linkage between groundwater quality and the processes of interest ( osmosis, and the associated groundwater level change) are discussed as a new result of the lithological evolution interpretation and its relation with the underground water circulation while the other indicators are introduced as a result of the classical statistical analysis methods (PCA).

Also, I think that it will be more significant as you said to organize the results and discussion tightly around each of the questions such that the readers are clear about how the scientific questions are addressed by the technical results. However, I may address your attention that at this stage of research, osmosis phenomenon and watertable static level rising are discussed as chemical and physical results of the combined effect of the water- rock contact. For fig. 5 in the line 92, I may inform you the map is done by a co-author when realizing this work.

For line 107. We can list first the processes that are investigated in the introduction. For processes listed as mineral dissolution, precipitation, reverse ions exchange and anthropogenic process I should inform you that these are the systematical result of the statistical method (correlation matrix) however osmosis is introduced as a new result as I said before. For Line 109. We can briefly clarify what "standardized" refers to. For section 2.3 we will clarify the rationale of employing WQI in this study and briefly clarify how WQI is calculated in this study. To give you more detail here I may draw you attention that WOI is used to evaluate the water quality and to relate it to the geographical context. This aims by the end to make a water management recommendations. Section 3. the measurements from the sampling analysis will be shown and the uncertainties of these measurements will be clarified. for Line 236. Osmosis phenomenon is not quantified at this stage of research as I said but it is introduced as a new result in the new concept introduced (Preferential dissolution corridors) and more detail will be introduced in the final correction. For line 240-256 I will remove generalities and add the reference results/figures that support the argument in the discussion. For Fig 12, I should draw your attention that this area is not a lake. In the past water did not appear and figure shows the abnormal static level of the watertable done by the lithological subsidence. Hope my explanation is useful. Best regards.